# Chalcogen bond-guided conformational isomerization enables catalytic dynamic kinetic resolution of sulfoxides

Jianjian Liu [1], Mali Zhou [1], Rui Deng [1], Pengcheng Zheng [1] & Yonggui Robin Chi [1,2]

Conformational isomerization can be guided by weak interactions such as chalcogen bonding (ChB) interactions. Here we report a catalytic strategy for asymmetric access to chiral sulfoxides by employing conformational isomerization and chalcogen bonding interactions. The reaction involves a sulfoxide bearing two aldehyde moieties as the substrate that, according to structural analysis and DFT calculations, exists as a racemic mixture due to the presence of an intramolecular chalcogen bond. This chalcogen bond formed between aldehyde (oxygen atom) and sulfoxide (sulfur atom), induces a conformational locking effect, thus making the symmetric sulfoxide as a racemate. In the presence of N–heterocyclic carbene (NHC) as catalyst, the aldehyde moiety activated by the chalcogen bond selectively reacts with an alcohol to afford the corresponding chiral sulfoxide products with excellent optical purities. This reaction involves a dynamic kinetic resolution (DKR) process enabled by conformational locking and facile isomerization by chalcogen bonding interactions.

Non–covalent interactions based on hydrogen bond[1–3] and halogen bond[4–7] represent a powerful and promising activation mode in catalytic synthesis. However, the chalcogen bond is a new class of weak non–covalent interactions between the chalcogen atom (S, Se, Te) and Lewis base (Fig. 1a), which attracted attentions only in recent years[8–10]. In the living systems, the chalcogen bonding interactions play a crucial role in regulating protein conformations[11] and preserving certain enzymatic activities[12,13] (Fig. 1b). These interactions have also been studied in the areas of solid–state chemistry[14], anion recognition[15–17], supramolecular assembling[18–20], and drug designs[21,22]. For example, the conformational locking effect induced by chalcogen bonds is believed to enhance the bioactivities of multiple commercial pharmaceuticals such as Acetazolamide[23] and Selenazofurin[24]. (Fig. 1b). In contrast to the relatively wide applications in functional molecule design, chalcogen bonds are much less explored as effective tools for catalysis and organic synthesis especially in asymmetrical reactions[25]. The use of

chalcogen bonding (ChB) for catalysis received reasonable attentions only in recent years[26,27]. As disclosed by Matile[28,29], Huber[30,31] and Wang[32–34], the key is to install chalcogen bond donors to the catalysts that can interact with the substrate for catalytic activations (Fig. 1c). Most of the success for effective catalysis comes from cationic chalcogen bonding interactions, which cationic charges are introduced to decrease the electron density of chalcogen atom to enhance chalcogen bonding interaction. Despite these impressive progresses, the development of effective chalcogen bonding catalysis remains slow, and evidences for the presence of chalcogen bond in catalytic reactions mostly relies on in situ NMR spectra ($^{13}$C, $^{77}$Se)[27,32–35], UV-vis and nanoESI-MS[15] analysis. We postulate that part of the reasons lie on the difficulties in designing these stable chalcogen bonded complex between catalysts and substrates.

We're particularly motivated by the fact that such intramolecular interactions are widely present (or can be readily installed) in both

[1]State Key Laboratory Breeding Base of Green Pesticide and Agricultural Bioengineering, Key Laboratory of Green Pesticide and Agricultural Bioengineering, Ministry of Education, Guizhou University, Guiyang 550025, China. [2]Division of Chemistry & Biological Chemistry, School of Physical & Mathematical Sciences, Nanyang Technological University, Singapore 637371, Singapore. ✉e-mail: pczheng@gzu.edu.cn; robinchi@ntu.edu.sg

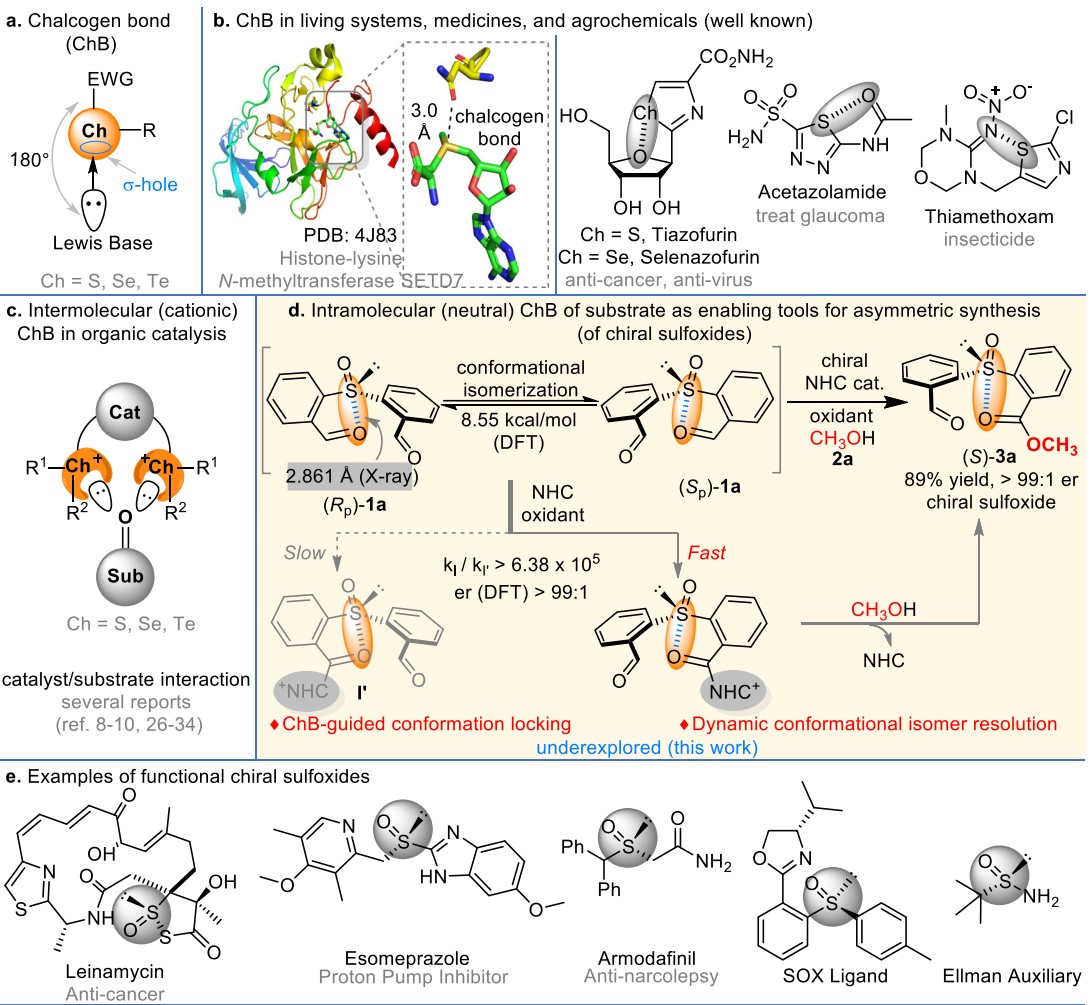

**Fig. 1 | Chalcogen bonding in functional molecules and asymmetric synthesis.**
**a** Chalcogen bond (ChB). **b** ChB in living systems, medicines, and agrochemicals.
**c** Intermolecular (cationic) ChB in organic catalysis. **d** Intramolecular (neutral) ChB
of substrate as enabling tools for asymmetric synthesis (of chiral sulfoxides).
**e** Examples of functional chiral sulfoxides.

macro[18–20] and small molecules of natural origins[11–13] or chemical synthesis[25–34,36]. It is also encouraging to observe that chalcogen bonding in the intramolecular fashion can be designed in readily predictable and modular manners[21–24]. For instance, Tomada et al reported a chiral selenenylation reagent bearing the intramolecular N-Se interaction to rigidify the whole molecule[37]. Subsequently, Wirth extended this concept towards O-Se interactions and achieved the asymmetric functionalization of alkenes[38]. Furthermore, the transient intramolecular chalcogen bonding interactions had been proven by Smith et al which were the crucial force in stereoselectivity control[25,39–44]. Based on these thought-provoking applications of intramolecular ChBs[45–48], our interests are directed toward employing intramolecular chalcogen bonding interactions for conformational regulations and selective chemical transformations.

In this work, we study the involvement of ChB interactions in the reactivity of sulfinyldibenzaldehyde compounds. These noncovalent interactions are key to achieve the regioselective mono-esterification of the compound via chiral carbene-catalyzed oxidation process, enabling the preparation of chiral sulfoxides. These results indicate that ChB interactions can play an important role in asymmetric organic synthesis.

## Results
### Reaction development
Here we disclosed a catalytic dynamic kinetic resolution protocol of sulfoxides which were enabled by intramolecular chalcogen

bond-guided conformational isomerization of the substrate (Fig. 1d). The sulfoxide group have been testified to be a ChB donor[8,49] and the solid-state X-ray structure of **1a** (Fig. 1d, details see Supplementary Table 2) suggests the presence of chalcogen bond between sulfur and oxygen atom (bond length = 2.861 Å). This chalcogen bond breaks the symmetry of **1a**, and therefore makes this symmetric sulfoxide present as a racemic mixture with two conformational enantiomers [($R_p$)-**1a** and ($S_p$)-**1a**, Fig. 1d]. Interconversion of the two enantiomers via chalcogen bond-guided conformational isomerization was estimated by DFT, which was a facile process with an activation energy of 8.55 kcal/mol. We postulated that the chalcogen bonding likely exist in solution as well, and then started to screen suitable conditions for a dynamic kinetic resolution of such conformers by using an NHC-catalyzed esterification process. Under the catalysis of N-heterocyclic carbene (NHC)[50–54] at an oxidative condition to covert one of the aldehyde moieties of **1a** to an ester, a highly efficient dynamic kinetic resolution of this sulfoxide (**1a**) is realized. Oxidation of Breslow intermediates to the corresponding acyl azolium intermediates (**I** and **I'**) were estimated (via DFT) to be the stereoselectivity-determining step in this DKR process. Our reaction affords chiral sulfoxide products with good yields and excellent enantiomeric purities. Notably, chiral sulfoxides are widely used in medicines (such as Esomeprazole[55] and Armodafinil[56]), agrochemicals (such as Ethiprole[57]), and as ligands in asymmetric catalysis[58,59] (Fig. 1e). The chiral sulfoxides from our reactions may

**Table 1 | Optimization of reaction conditions[a]**

| Entry | NHC | base | solvent | yield (%)[b] | er[c] |
|---|---|---|---|---|---|
| 1 | A | $K_2CO_3$ | THF | 45 | 99:1 |
| 2 | B | $K_2CO_3$ | THF | 52 | 99:1 |
| 3 | C | $K_2CO_3$ | THF | 55 | 98:2 |
| 4 | D | $K_2CO_3$ | THF | 44 | 92:8 |
| 5 | B | $Na_2CO_3$ | THF | 56 | 98:2 |
| 6 | B | $Cs_2CO_3$ | THF | 35 | 99:1 |
| 7 | B | $K_3PO_4$ | THF | 85 | 97:3 |
| 8 | B | DBU | THF | 76 | 94:6 |
| 9 | B | $Et_3N$ | THF | 82 | 96:4 |
| 10 | B | $K_3PO_4$ | $CH_2Cl_2$ | 89 | >99:1 |
| 11 | B | $K_3PO_4$ | EtOAc | 75 | 98:2 |
| 12 | B | $K_3PO_4$ | toluene | 50 | 99:1 |

[a]Unless otherwise specified, the reactions were carried under $N_2$ atmosphere using **1a** (0.10 mmol), DQ (0.10 mmol), $CH_3OH$ (0.12 mmol), pre–NHC (0.01 mmol), base (0.02 mmol), solvent (2.0 mL), 30 °C, 12 h. [b]Isolated yield of **3a**. [c]The er values of **3a** were determined via HPLC on the chiral stationary phase.

work as platform scaffolds for transforming to bioactive molecules and catalysts.

At first, we chose conformational isomeric sulfinyldibenzaldehyde **1a** as the model sulfoxide substrate and methanol **2a** as a nucleophile to search for suitable conditions, and the key results were summarized in Table 1. Triazoliums were explored as the NHC pre–catalysts with diphenoquinone (DQ)[60] as an oxidant to convert one of the aldehyde moieties of **1a** to an ester unit. An encouraging result was obtained when aminoindanol–derived triazoium **A** was the NHC pre–catalyst with $K_2CO_3$ as a base in THF, offering the corresponding chiral sulfoxide product **3a** in 45% yield and 99:1 er (entry 1). Replacing the counter anion ($BF_4^-$) in **A** with a chloride ion (pre–catalyst **B**) led to comparable results for this model substrate (entry 2). As an important technical note, in subsequent studies for scope explorations, we found that pre–catalyst **B** consistently performed better for all the substrate examinations. The N–mesityl substituent in **A** could be switched by a phenyl unit (pre–catalyst **C**) with little effect on product yield or er value (entry 3). Further optimizations with respects to bases and solvents were performed by using NHC pre–catalyst **B** (entries 5–12). At last, we found that by using $K_3PO_4$ as the base with $CH_2Cl_2$ as the solvent, product **3a** could be isolated in 89% yield with over 99:1 er (entry 10).

## Substrate scope

Having an acceptable condition in hand, the generality of the reaction was then investigated (Fig. 2). Various substituents were placed on the para–carbon (relative to the aldehyde moiety) on the phenyl ring of **1a**, in all cases the mono–ester products were obtained with excellent er values (mostly over 99:1 er, **3b** to **3j**). The reaction yields are good as well when the substituents are methyl (**3b**), methoxyl (**3c**), ethylthio (**3d**) or halogen atoms (**3e–3g**), giving the corresponding products with 60–94% yields. When electron–withdrawing units (e.g., CN, $CF_3$) were used, the products (**3 h, 3i**) were obtained in slightly lower yields (60% and 61% yields) with excellent er values maintained. The main side products were from further esterification reaction of **3h** and **3i** to

give the corresponding di–ester adducts. Various substituents (such as Me, OBn and halogen) could be installed on the meta–carbon (relative to the aldehyde) on the phenyl ring of **1a** as well without affecting reaction yields and er values (**3k–3o**). Remarkably, substrates with two substituents on both the para– and meta–carbons of **1a** were well tolerated (**3p** and **3q**). When a methyl unit was placed on the ortho–position (relative to aldehyde) of **1a**, drops on both reaction yield and er value were observed (**3s**). The low yield of **3s** was mainly due to di–ester formation, and the origins for the decrease of er value may result from steric hindrance. Fluorine substituent at ortho–position led to product **3t** with 80% yield and 95:5 er. Placing a methyl unit on the ortho–carbon (relative to the sulfoxide unit) of **1a** led to **3r** with over 99:1 er, albeit with a decreased 47% yield. Moreover, various alcohols and thiols, including secondary alcohols, could also be used as effective nucleophiles to replace methanol (**3u–3x**). Interestingly, when diols were used as the nucleophiles, both of hydroxyl moieties could be acylated to give the corresponding chiral di–sulfoxides with excellent yields and er values (**3y, 3z**). These results suggested that our strategy may be further developed to attach chiral sulfoxide to functional molecules (such as natural products and polymers) which contain multiple hydroxyl units.

## Synthetic transformations

In synthetic applications, our approach could be readily scaled up to 1.2 grams only with little influence on product yield (e.g., **3a**, 1.2 grams, 79% yield, and >99:1 er; Fig. 2). The remaining aldehyde unit in our sulfoxide product **3a** could be easily converted to a diverse set of functional groups (Fig. 3a). For instance, the hydrogen of aldehyde could be deuterated[61] catalyzed by achiral NHC in the presence of $D_2O$ to afford 100% deuterated **4a** in 77% yield and without the loss of optical purity. Moreover, the formyl group could be cyanation[62] and thioesterification catalyzed by achiral NHC with high er values (**4b, 4c**). Enantioenriched terminal alkyne **4d** and alkene **4e** were synthesized efficiently by means of Seyferth–Gilbert reaction[63] and Wittig reaction[64], respectively. Chiral

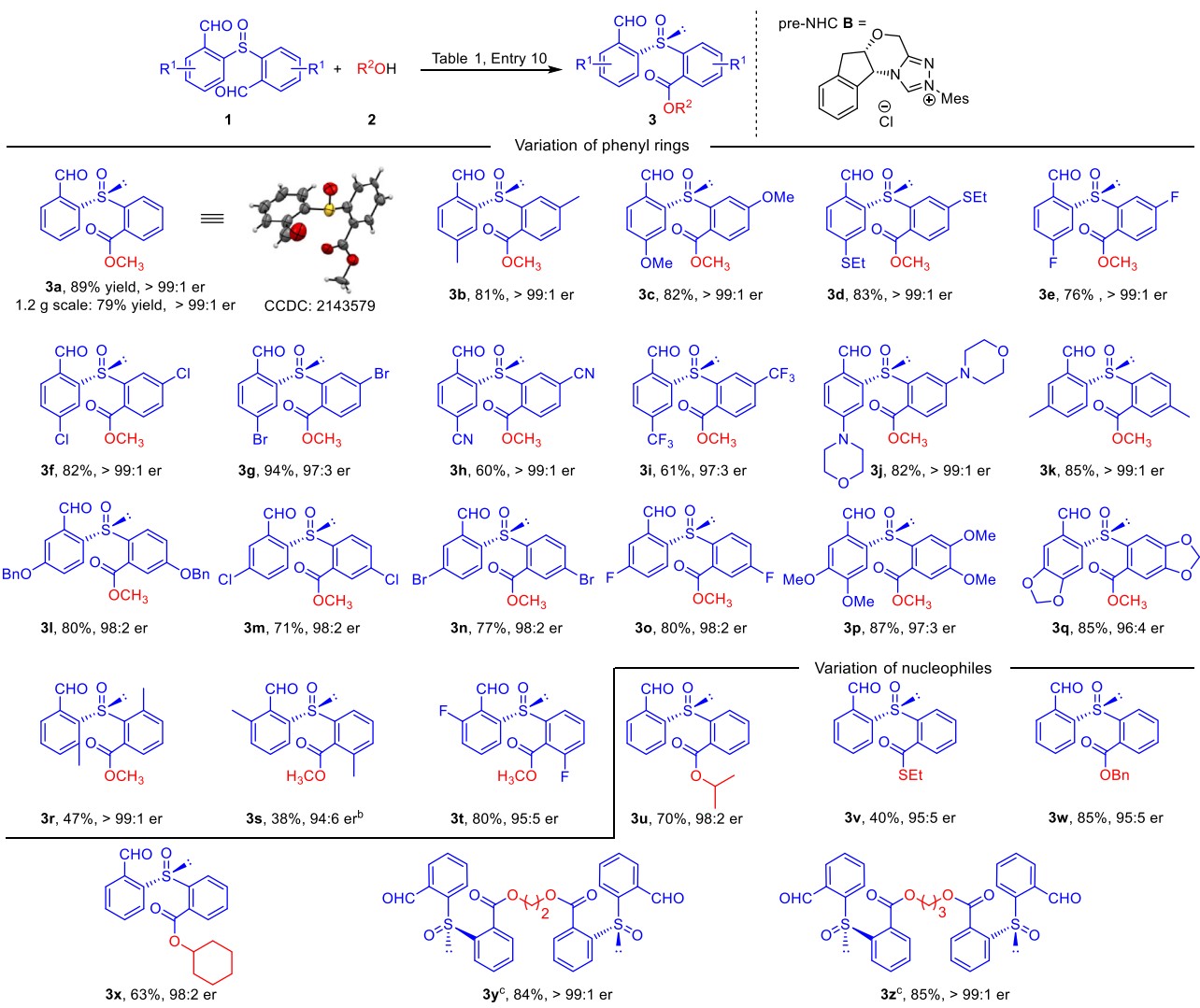

**Fig. 2 | Substrates scope of the reaction[a].** [a]Reaction conditions as stated in Table 1, entry 10. Yields are isolated yields after purification by column chromatography. Er values were determined via HPLC on chiral stationary phase. [b]50 °C and THF as solvent. [c]220 mol% **1a**, DQ, K₃PO₄ and 100 mol% diol were used.

sulfoxide **3a** reacted with L–valinol[65] generated oxazolines **4g** was very similar to the SOX type ligands[66,67] (Fig. 1e). Noteworthily, the chiral sulfoxide **4h** and its analogues have been proven as a chiral ligand and catalyst in several asymmetric synthesis[59]. It could be easily synthesized from **3a** via reductive amination reaction as well as its analogues. Furthermore, **3a** underwent hydrolysis of the ester group and subsequent reductive amination of the formyl group with BnNH₂ to afford an unnatural amino acid **4i** bearing a chiral sulfoxide center with good yield and excellent er value. Combination of **3a** with Ellman auxiliary[68] accessed to a chiral disulfoxide product **4j** efficiently via a concise condensation reaction with 90% yield.

Moreover, two practical applications of **4d** and **4g** were verified. Alkyne **4d** could be conjugated with an anti–HIV drug (Zidovudine)[69] which possessing an azido group to afford a modified Zidovudine **5a** with moderate yield. As we expected, **4g** could be a potential chiral ligand in asymmetric synthetic chemistry, which was used as a chiral ligand in the Pd-catalyzed enantioselective substitution reaction[70] between the alkene **6** and the malonate **7**, with the chiral product **8** afforded in 98:2 er. (Fig. 3b)

## Mechanistic studies

To understand the possible impacts of chalcogen bonding interactions, we examined two other sulfoxide substrates (**1aa** and **1ab**) by placing the positions of aldehyde moieties which are far away from the sulfoxides sulfur center (Fig. 4a). From analysis on the X–ray structure of **1aa**, the remote aldehyde unit does not show any chalcogen bonddind interaction with the sulfur atom (see Supplementary Table 2). It is therefore expected that the chalcogen bond–guided conformational resolution strategy developed here shall not work for substrates such as **1aa** and **1ab**. This expectation was verified by our experimental observations when the use of **1aa** and **1ab** under our condition, It gave the corresponding products (**3aa** and **3ac**) with nearly no enantiomeric excesses and the yields of di-esters were increased.

To probe further mechansic insights of our reactions, we investigated the chalcogen bonding strength between the sulfoxide and formyl group by using DFT calcuations (Fig. 4b). The structure provided by single crystal diffraction data of **1a** was used as the initial point for geometric optimizations. The chalcogen bond energy (ChBE) was estimated to be 3.44 kcal/mol (Fig. 4b). Furthemore, in order to evalued the influence of substituents to the ChBs, substrates **1c** (with OMe) and **1i** (with CF₃) were examined with the same DFT calculation method. The initial structures for DFT calculations were obtained from the corresponding single crystal of **1c** (CCDC **2172904**) and **1i** (CCDC **2172911**). The results showed that the chalcogen bond strength of **1c** and **1i** were 3.56 and 4.27 kcal/mol, respectively (see Supplementary Fig. 1 for details). Moreover, the additions of NHC catalyst to aldehyde

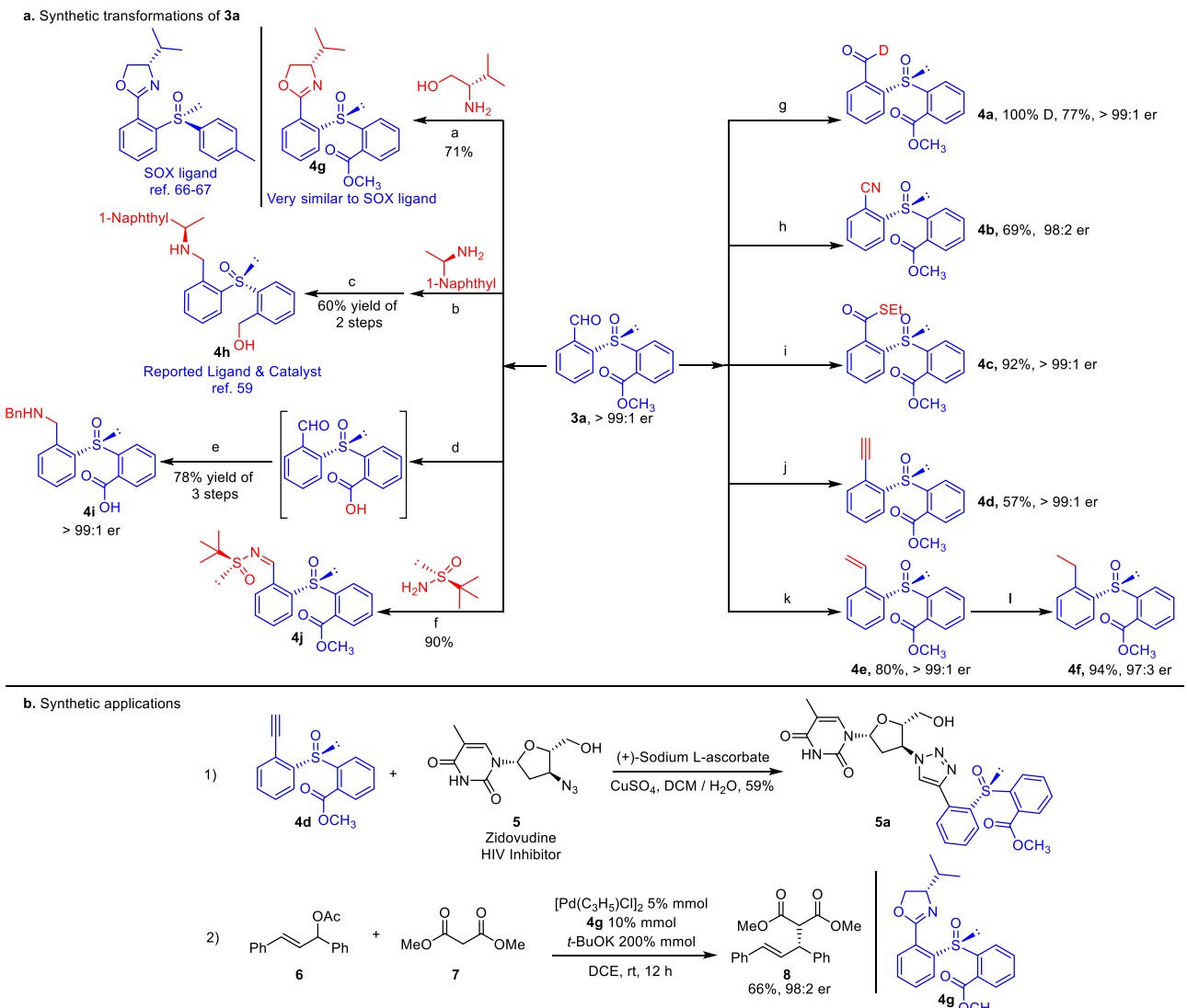

**Fig. 3 | Synthetic transformations and applications.** [a]K₃PO₄, NBS, 4 Å MS, 30 °C, toluene; [b]MgSO₄, 4 h, CH₂Cl₂, NaBH₃CN; [c]NaBH₄, Ti(OEt)₄, CH₂Cl₂; [d]LiOH, THF:H₂O = 2:1, 2 h, then 1 M HCl; [e]MgSO₄, BnNH₂, 4 h, then NaBH₃CN; [f]Pyrrolidine, 4 Å MS, CH₂Cl₂, 60 °C; [g]achiral NHC, AcOK, D₂O:CH₂Cl₂ = 4:1; [h]achiral NHC, TsNH₂, Et₂NH, 4 Å MS, toluene; [i]achiral NHC, DQ, EtSH, K₃PO₄, CH₂Cl₂; [j]TMSCHN₂, LDA, THF, −78 °C; [k]CH₃PPh₃Br, KHMDS, THF; [l]Pd/C, H₂, EtOH. **a** Synthetic transformation of 3a. **b** Synthetic applications.

moiety in the two conformational isomers (two set of enantiomers; four possibilities for the additions) were then evaluated (Fig. 4c). We found that the aldehyde moieties involved in chalcogen bonding interactions are conformationally locked and weakly activated. These conformationally locked aldehyde moieties react faster with the NHC catalyst ($\Delta G^{\ddagger}$ = 5.50, 3.53 kcal/mol) than the no chalcogen bonded aldehyde moieties ($\Delta G^{\ddagger}$ = 6.49, 9.24 kcal/mol). Meantime, the low rotation barriers (Fig. 4b) indicates that the conformations of 1a can undergo rapid interconversions at room temperature, making it feasible to achieve a carbene–catalyzed DKR process. Further DFT studies suggest that oxidation of the Breslow intermediate (Fig. 4d) is the stereo–determine step. The activation energy difference of **Ox–Ts I** and **Ox–Ts I′** ($\Delta G^{\ddagger}$ = 14.16, 22.22 kcal/mol, respectively) is estimated as 8.06 kcal/mol, suggesting an er value over 99:1, that is consistent with our experimental observations (see Supplementary Fig. 3 for details).

In summary, we have disclosed a carbene–catalyzed DKR strategy for the synthesis of chiral sulfoxides. This method takes advantage of intramolecular chalcogen bonds installed in molecules to guide conformational isomerization and reactivity differentiation of substrates. In particular, through a chalcogen bonding–enabled reactivity

differentiation, we realize a carbene–catalyzed dynamic kinetic resolution process for efficient preparation of chiral sulfoxides with excellent optical purities. The chiral sulfoxide products from our reactions may serve as platform scaffolds for straightforward transformation to useful molecules with applications in catalysis and biological studies. Chalcogen bonding interactions are naturally present or can be readily installed in various molecules. The strategy reported herein may open a new avenue in reaction control and asymmetric synthesis.

## Methods
### General procedure for the catalytic reactions
To a 100.0 mL over–dried round bottom flask equipped with a magnetic stir bar was added **1a** (1.0 g, 3.87 mmol), DQ (1.58 g, 3.87 mmol), pre–NHC **B** (139.8 mg, 0.38 mmol) and K₃PO₄ (164.2 mg, 0.77 mmol). The flask was then sealed, purged and backfilled with N₂ three times in glovebox before adding CH₂Cl₂ (60.0 mL) and CH₃OH (0.19 mL, 4.65 mmol), and the reaction mixture was stirred in oil bath at 30 °C for 12 h. The mixture was concentrated under reduced pressure. The resulting crude residue was purified via column chromatography on

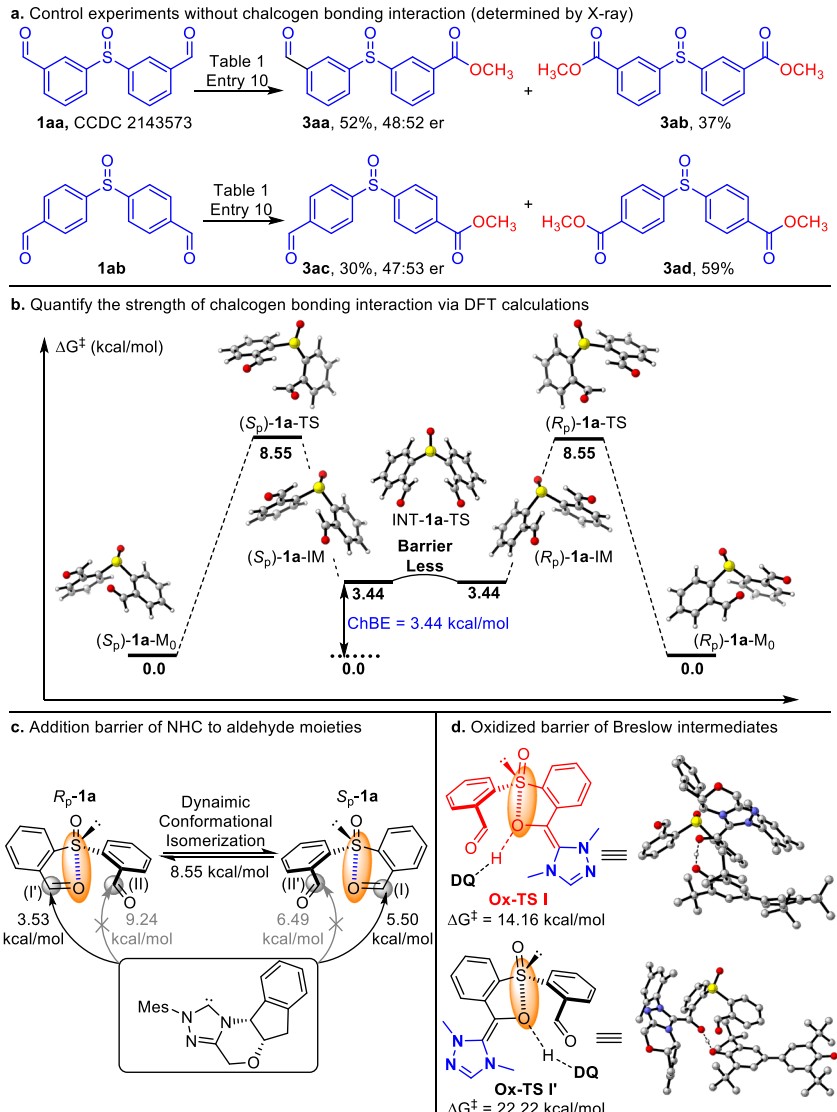

**Fig. 4 | Mechanistic studies. a** Control experiments without chalcogen bonding interaction. **b** Quantify the strength of chalcogen bonding interaction via DFT calculations. **c** Addition barrier of NHC to aldehyde moieties. **d** Oxidized barrier of Breslow intermediates.

silica gel by using petroleum ether / ethyl acetate (2:1) to afford the desired product **3a** (881.8 mg, 79% yield, > 99:1 er).

## Data availability

The experimental method and data generated in this study are provided in the Supplementary Information file. Geometries of all DFT-optimized structures (in.xyz format) are provided as Supplementary Data file. The crystallographic data for structures of **1a**, **1c**, **1i**, **1aa** and **3a** have been deposited in the Cambridge Crystallographic Data Centre under accession CCDC code **2143570**, **2172904**, **2172911**, **2143573** and **2143579**, respectively. Copies of the data can be obtained free of charge via www.ccdc.cam.ac.uk/data_request/cif.

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

## Acknowledgements

We acknowledge funding supports from National Natural Science Foundation of China (21732002, 22061007, P. C. Z; 22071036, Y. R. C); Frontiers Science Center for Asymmetric Synthesis and Medicinal Molecules, Department of Education, Guizhou Province [Qianjiaohe KY number (2020)004, Y. R. C]; The 10 Talent Plan (Shicengci) of Guizhou Province ([2016]5649, Y. R. C); Qiankehejichu-ZK[2022]zhongdian024, P. C. Z); Science and Technology Department of Guizhou Province ([2018]2802, [2019]1020, Y. R. C); Program of Introducing Talents of Discipline to Universities of China (111 Program, D20023, Y. R. C) at Guizhou University. Singapore National Research Foundation under its NRF Investigatorship (NRF-NRFI2016–06, Y. R. C) and Competitive Research Program (NRF-CRP22–2019–0002, Y. R. C); Ministry of Education, Singapore, under its MOE AcRF Tier 1 Award (RG7/20, RG5/19, Y. R. C), MOE AcRF Tier 2 (MOE2019-T2–2–117, Y. R. C), and MOE AcRF Tier 3 Award (MOE2018–T3–1–003, Y. R. C), Nanyang Technological University.

## Author contributions

Y.R.C and P.C.Z conceptualized and directed this research; J.J.L designed and performed main methodology development, scope evaluation and synthetic application; M.L.Z and R.D synthesized the substrates; P.C.Z conducted the DFT calculations. All authors contributed to discussions and manuscript preparation.

## Competing interests

The authors declare no competing interests.
