## [Peer Review File · Nature Communications]

Chalcogen Bond-Guided Conformational Isomerization Enables Catalytic Dynamic Kinetic Resolution of SulfoxidesREVIEWER COMMENTS

Reviewer #1 (Remarks to the Author):

The authors present a dynamic kinetic resolution protocol towards the synthesis of enantiopure sulfoxides, which is based on intramolecular chalcogen bonding and subsequent asymmetric functionalization of one of two aldehyde moieties. The target compounds are obviously of relevance in biological/pharmaceutical context, and the method presented here features a broad substrate scope. Most importantly, it constitutes, to the best of my knowledge, the first realization of such a concept involving chalcogen bonding, and thus adds a clever twist to the emerging use of this interaction. Overall, I would regard the novelty as suitable for publication in Nature Communications, and I find the conclusions drawn here all plausible. Consequently, I recommend publication once some minor issues have been taken into account:

#1. The authors state that "evidence for the presence of ChB in catalytic reactions are from NMR spectroscopic analysis" and it sounds like this is the only evidence. This not true however, as multiple authors have also used reference compounds and other means to point towards the relevance of ChB for the mode of activation.

#2. I find the citation of previous literature in many cases to be severely lacking. This most importantly refers to two aspects: a) the authors state that "... intramolecular interactions are widely present ... in chemical synthesis", then cite a series of papers which are all very relevant to this study, but only very partially contain intramolecular ChBs; b) in the mid-1990s, both Tomoda and Wirth have used intramolecular ChB to rigidify chiral selenylating agents and since these pioneering work bears some relevance to the current study (use of intramolecular ChB in stereoselective synthesis), I think they should be cited, even though the present approach is of course conceptually different and innovative. The corresponding papers can be found in virtually any review on ChBs.

Reviewer #2 (Remarks to the Author):

The manuscript by Y. R. Chi et al. presents experimental and computational evidence of a conformational isomerization guided by chalcogen bonding as an strategy for asymmetric synthesis of chiral sulfoxides. The idea of using this conformational locking is clever and the chemistry has been competently done. Perhaps the mechanistic studies could be more complete by studying how substitution effects affect the capability of the carbonyl group as chalcogen bond acceptor. It would be also interesting the computational exploration of other chemical groups acting as Lewis bases instead of the carbonyl present in 1a.

In summary, I consider the present results are novel enough to be published in Nat. Commun. after addressing a few minor points:

- A recent work on the nature and strength of carbonyl...sulfoxide interactions (Cryst. Growth Des. 2021, 21, 4, 2481–2487) could be interesting for the authors and should be cited here.
- Has the effect of the solvent been taken into account in the DFT calculations?
- English language could be improved, particularly in the introduction.

Point-by-point response to the reviewers' comments

Response to referee #1:

Referee #1 considers that our study describes “not only the target compounds are obviously of relevance in biological/pharmaceutical context, but also the method presented here features a broad substrate scope and thus adds a clever twist to the emerging use of this interaction”. We truly appreciate the very thoughtful and detailed questions raised by this referee, and have taken the possible efforts to address these comments, as detailed below :

#1 The authors state that "evidence for the presence of ChB in catalytic reactions are from NMR spectroscopic analysis" and it sounds like this is the only evidence. This not true however, as multiple authors have also used reference compounds and other means to point towards the relevance of ChB for the mode of activation.

Our Response: Revised.

#2 I find the citation of previous literature in many cases to be severely lacking. This most importantly refers to two aspects: a) the authors state that "... intramolecular interactions are widely present ... in chemical synthesis", then cite a series of papers which are all very relevant to this study, but only very partially contain intramolecular ChBs; b) in the mid-1990s, both Tomoda and Wirth have used intramolecular ChB to rigidify chiral selenylating agents and since these pioneering work bears some relevance to the current study (use of intramolecular ChB in stereoselective synthesis), I think they should be cited, even though the present approach is of course conceptually different and innovative. The corresponding papers can be found in virtually any review on ChBs.

Our Response: Revised. These corresponding literatures have been updated in manuscript (ref 37-48).

Response to referee #2:

Referee #2 is quite positive and point out that our studies are novel enough to be published in Nat. Commun. after addressing a few minor points. His thought-provoking comments mainly deal with the influence of substituents Lewis base and solvent to the ChB. We have taken the best efforts to address these comments, as detailed below :

#1 Perhaps the mechanistic studies could be more complete by studying how substituent effects affect the capability of the carbonyl group as chalcogen bond acceptor. It would be also interesting the computational exploration of other chemical groups acting as Lewis bases instead of the carbonyl present in 1a.

Our Response: To further investigate the substituent effects, more additional DFT calculations have been performed. We grew some new crystals of substrates (**1c**, **1i** and **1ac**) to obtain the exact the structural details for DFT calculations. Finally, it was found that the different type substituents will influence the strength of chalcogen bond (ChB). Generally, the dialdehyde substrate bearing electron donating group (OMe, **1c**) will weaken the strength of ChB, and the ChB will be enhanced by bearing electron withdrawing group (CF₃, **1i**). The aldehyde moieties were replaced by hydroxyl group, the ChB interaction can be observed in single crystal of **1ac**, and the strength of ChB was weaker than aldehyde moieties by DFT calculation. The new results have been updated in manuscript, for more details please see SI (page S19).

#2 A recent work on the nature and strength of carbonyl...sulfoxide interactions (Cryst. Growth Des. 2021, 21, 4, 2481–2487) could be interesting for the authors and should be cited here.

Our Response: Updated (ref 49).

#3 Has the effect of the solvent been taken into account in the DFT calculations?

Our Response: In the DFT calculation, the polarizable continuum model using the integral equation formalism variant (IEFPCM) was employed to optimization the geometry of substrates and transition states. More details have been attached in SI (page 18).

#4 English language could be improved, particularly in the introduction.

Our Response: Revised.

REVIEWERS' COMMENTS

Reviewer #2 (Remarks to the Author):

The authors have addresses all the comments successfully and the manuscript is ready to be published.